# Extracts of Fruits and Plants Cultivated In Vitro of *Aristotelia chilensis (Mol.) Stuntz* Show Inhibitory Activity of Aldose Reductase and Pancreatic Alpha-Amylase Enzymes

**DOI:** 10.3390/plants11202772

**Published:** 2022-10-20

**Authors:** Adriana Pineda, Andrea Arenas, Juan Balmaceda, Gustavo E. Zúñiga

**Affiliations:** 1Laboratorio de Fisiología y Biotecnología Vegetal, Departamento de Biología, Facultad de Química y Biología, Universidad de Santiago de Chile, Santiago 917021, Chile; 2Centro para el Desarrollo de la Nanociencia y la Nanotecnología (CEDENNA), Santiago 917021, Chile

**Keywords:** maqui berry, in vitro culture, antioxidants, aldose reductase, pancreatic alpha-amylase, enzyme inhibitor, phenolic compounds

## Abstract

*Aristotelia chilensis* is a plant whose fruit is considered a powerful natural antioxidant. During the last years, some investigations of the fruit have been carried out, finding antioxidant properties in the juice or the phenolic fraction. The antioxidant properties of the plant are useful in the inhibition of enzymes related to diabetes such as pancreatic aldose reductase and alpha-amylase. Because many synthetic drugs used today have limitations and potentially harmful side effects, the use of naturally occurring compounds, such as flavonoids, is clinically attractive. In this study, the characterization of aqueous extracts of fruits and in vitro plants of *A. chilensis* was carried out based on their content of anthocyanins and total phenols, the antioxidant capacity by the antiradical activity 2,2-diphenyl-1-picrilhydrazil (DPPH), and the profile of anthocyanins and other phenolic compounds by liquid chromatography coupled to mass spectrometry (LC-MS/MS). Subsequently, the effect of these extracts on the inhibition of bovine aldose reductase and pancreatic alpha-amylase enzymes was determined. According to our results, extracts of fruits and in vitro plants of *A. chilensis* achieved inhibition of the bovine aldose reductase enzyme of 85.54 ± 1.86% and 75.67 ± 1.21%, respectively. Likewise, the percentage of inhibition of the pancreatic alpha-amylase enzyme for fruit extracts was 29.64 ± 0.63%, while for in vitro plant extracts it was 47.66 ± 0.66%. The antioxidant and enzymatic inhibition activity of the extracts were related to the content of anthocyanins, such as delphinidin and cyanidin glycosides as well as the phenols derived from quercetin, myricetin, and kaempferol. The results obtained allow us to suggest that the in vitro culture of plants of *A. chilensis* represents a viable biotechnological alternative to obtain phenolic compounds for the inhibition of aldose reductase and pancreatic alpha-amylase enzymes.

## 1. Introduction

Diabetes mellitus is a chronic disorder of the metabolism of carbohydrates, lipids, and proteins, with type II diabetes being the most common form, characterized by insulin resistance and an increase in the glycemic index immediately after eating (postprandial hyperglycemia) [1,2]. According to the World Health Organization, in 2008 there were around 347 million people with diabetes and current estimates suggest that this number will double by 2030 due to changes in lifestyle and consumption of high carbohydrate diets [3,4].

Currently, there are several therapeutic approaches to the treatment of diabetic complications, including the inhibition of enzymes such as aldose reductase (EC 1.1.1.21) and alpha-amylase (EC 3.2.1.1). The first one is a Nicotinamide Adenine Dinucleotide Phosphate (NADPH) oxidoreductase, monomeric, cytosolic, and NADPH-dependent enzyme, which catalyzes a reduction in a variety of aldehydes and carbonyls in the polyol metabolic pathway in which glucose is reduced to sorbitol by consuming NADPH. However, sorbitol does not diffuse easily through cell membranes and its intracellular accumulation has been implicated in chronic diabetes complications [5,6], since it generates hyperosmolarity, which leads to an alteration in the plasma membrane permeability [4]. It generates long-term microvascular damage such as diabetic retinopathy characterized by a spectrum of lesions within the retina, which include changes in vascular permeability, capillary microaneurysms, and neovascularization. Almost all patients with type 1 diabetes, and most having type 2 diabetes, exhibit some retinal lesions after 20 years of disease [4]. On the other hand, alpha-amylase is a hydrolase enzyme that catalyzes the hydrolysis of internal α-1,4-glycosidic linkages in starch to yield products such as glucose and maltose that generate an increase in the postprandial glycemic index [7], which represents a therapeutic target of type II diabetes mellitus.

Several pharmacological strategies are used in the inhibition of both enzymes, however, many of the existing synthetic drugs have limitations and potentially harmful side effects related to poor tissue penetration and cell damage [3,8]. Due to this, the use of compounds of natural origin is clinically attractive. Some of the natural compounds with proven inhibitory activity of both enzymes are flavonoids, which are water-soluble plant molecules that have 15 carbon atoms and consist of 2 aromatic rings linked by a 3-carbon chain that forms an oxygenated heterocyclic ring [9]. These molecules have generated considerable interest due to their beneficial effects on human health such as antioxidant, antiviral, anti-inflammatory and anti-tumor activity [9,10].

*Aristotelia chilensis*, also known as maqui, is one of the plants of the Chilean native flora recognized for presenting high content of phenolic compounds. This tree of the Elaeocarpaceae family grows between the IV to the XI Region [11,12] and its fruits and leaves have been ethnobotanically used to treat diabetes, sore throat, kidney, digestive system, fever, migraines, injuries, etc., as well as in food processing and textile dyeing [13,14,15,16].

The use of maqui fruits to obtain bioactive extracts is limited by the availability of the raw material. Additionally, the composition varies according to the place where the fruits are collected. An evaluated strategy to ensure the production of plant material is in vitro tissue culture because this tool allows production of plant tissues with a high level of homogeneity due to the culture conditions [17].

One of the first reports of the in vitro culture of *A. chilensis* was Céspedes [18], which used 0.05–0.5 mg/L of Zeatin and 0.5–1.0 mg/L of 2,4-dichlorophenoxyacetic acid (2,4-D), obtained roots by indirect organogenesis. Moreover, Díaz [19] cultivated maqui callus in the presence of light, in media supplemented with 2.0 mg/L of 2,4-D, obtaining anthocyanin-producing cells. Likewise, Sadino [20] obtained pigmented callus in media with 3.0 mg/L of 2,4-D and 1.0 mg/L of kinetin (KIN).

In our search for the bioactive principles of plants, we evaluated the extracts of aqueous infusions of fruits and in vitro plants of *A. chilensis*, particularly in relation to their antioxidant activity and inhibitory effects on bovine aldose reductase and pancreatic alpha-amylase enzymes.

## 2. Results

In this study, we compared the biological activity of fruits and in vitro plants of *A. chilensis* (Figure 1). Aqueous fruit extracts had a higher total anthocyanin content (7.63 ± 0.09 mg delfinidine-3-glu/g DW) and total phenols (79.90 ± 3.77 mg AGE/g DW) compared with in vitro plant extracts, whose anthocyanin content was 0.40 ± 0.01 cyanidine-3-glu/g DW and total phenols of 19.65 ± 2.06 mg AGE/g DW). Likewise, there were no significant differences regarding the antioxidant activity of 2,2-diphenyl-1-picrylhydrazyl (DPPH) radical obtained in both extracts (DPPH IC_50_ fruit 0.11 ± 0.01 mg/g DW and DPPH IC_50_ in vitro plants 0.12 ± 0.01 mg/g DW) (Table 1).

In addition, the determination of the profile of anthocyanins and phenolic compounds present in both aqueous extracts was made by the comparison of the MS/MS fragmentation patterns reported in the literature [21], using as reference the available data for the maqui fruit and other berries, with which a tentative identification of such molecules was made.

In this study, a total of 20 anthocyanins were identified in the extracts obtained from fruits (Table 2). Of these, the ones with the highest relative abundance were: Delphinidin 3,5-diglucoside and Delphinidin-3-(feruloyl)-5-diglucosise. Figure 2 shows all those molecules that presented a relative abundance greater than 5%, which in total add up to 78.51%.

By contrast, in vitro plant extracts accumulated a total of 15 compounds (Table 3), of which those with the highest relative abundance were: Pelargonidin-3-glucoside (10.27%), Delphinidin 3-glucoside (14.32%), and Delphinidin-3-(6-feruloyl)-5-diglucoside (52.24%). Of the total molecules detected, only three compounds were exclusively identified in in vitro plants (Delphinidin 3-(2″-galloyl-6″-acetyl-beta-galactopyranoside) *m*/*z* 657.1; Pelargonidin 3-3″,6″-dimalonylglucoside *m*/*z* 604, and Delphinidin 3-glucoside m/z 464.2) (Figure 3).

The compound Delphinidin-3-(6-feruloyl)-5-diglucoside (m/z 803.1) was the one that presented the highest relative abundance in both extracts (26.95% fruits and 52.24% plants in vitro) and produced a fragment single MS2 at *m*/*z* 303 corresponding to the aglycone delphinidin, and the loss of two fragments at m/z 162 indicated the separation of two hexose residues, which by the retention time at which they eluted, would correspond to glucose; likewise, a residue of *m*/*z* 176 remained that would correspond to ferulic acid. The anthocyanin Delphinidin 3,5-diglucoside (*m*/*z* 627.1) was the second compound with the highest relative abundance in fruit extracts (20.53%), followed by Cyanidin 3-(3′′,6′′-dimalonylglucoside) (*m*/*z* 621.1) with 9.71%.

The analysis of phenolic compounds presents in the extracts of fruits and plants in vitro showed that 13 compounds were accumulated in fruits, and the ones with the highest relative abundance were Quercetin 4′-galactoside 3.5 dichlorogenic acid and Kaempferol-7-glucoside (Table 4, Figure 4).

In in vitro plants, 10 compounds were accumulated consisting of the ones with the highest relative abundance: Quercetin 4′-galactoside (14.94%), Rhamnetin (12.16%, Granatin B (11.17%), and Kaempferol 3-(4″,6″-diacetylglucoside)-7 rhamnoside (35.38%) (Table 5, Figure 5). In Appendix A are the chromatograms in positive and negative mode.

Once the profile of anthocyanins and phenolic compounds and the antioxidant activity of the fruit extracts and in vitro plants of *A. chilensis* were established, their activity was evaluated as inhibitors of the enzymes bovine aldose reductase and pancreatic alpha-amylase.

The aqueous extracts of the fruit inhibited the activity of the bovine aldose reductase enzyme by 85.54% ± 1.86%, while those of in vitro plants reached an inhibition of 75.67% ± 1.21%; likewise, the fruit extracts inhibited the pancreatic alpha-amylase enzyme by 29.64% ± 0.63%; while in vitro plants inhibited it by 47.66% ± 0.66% (Table 6). However, as mentioned above, the anthocyanin content of the fruit extracts and the total phenol content was approximately 19 and 4 times higher than of in vitro plant extracts, respectively (Table 1).

## 3. Discussion

The characterization of the antioxidant activity of the aqueous extracts of *A. chilensis* shows that although the in vitro plant extracts had a phenolic content 4 times lower and an anthocyanin content almost 20 times less than quantified in the fruit extracts, there were no statistically significant differences in the antioxidant activity of the DPPH radical of both extracts, which suggests that the compounds present in in vitro plants could have higher antioxidant activity and that these molecules would correspond mainly to phenolic compounds.

The antioxidant capacity of the plant extracts evaluated is especially relevant when taking into account the oxidative stress that results from the increase in the products and the reaction of the enzymes (α-glucosidase and α-amylase) related to diabetes mellitus. Like other diseases, diabetes has been linked to free radical generation, with glucose autoxidation being a major source of free radicals in chronic hyperglycemia [22].

When evaluating the activity of the aqueous extract of the fruit, an enzymatic inhibition of 85.54% for aldose reductase and 29.64% for pancreatic alpha-amylase was achieved, while the in vitro plant extract reached 75.67% and 47.66% inhibition, respectively. Taking into consideration that these enzymes in normoglycemic conditions contribute to the control of oxidative stress through the detoxification of the main products of lipid peroxidation of biological membranes [4,23], this is a favorable result, since these extracts could have an effect on the control of diabetes.

From the biotechnological point of view, the most stable secondary metabolites could be obtained by using in vitro tissue culture techniques since, with the use of growth regulators, continuous plant production could be generated under highly regulated abiotic factors. This would allow the compounds of interest to be obtained from other tissues of the plant such as corms, making it possible to shorten the time required for production because fruits are obtained annually and depend largely on biotic factors such as pollinating agents, predators, and phytopathogens. Additionally, a more sustainable management of plant resources would be achieved with this approach.

The antioxidant activity of the extracts can be attributed to the presence of compounds that can donate electrons to free radicals with unpaired electrons by two mechanisms. The first corresponds to the attack of the hydroxyl group of ring B of the anthocyanin structure and the second corresponds to the attack of the oxonium ion in the C ring [24]. In general, such antioxidant activity is associated with the number of free hydroxyls around the ring of the molecule, where the greater the number of hydroxyls, the greater the antioxidant activity. Anthocyanins with their 3′, 4′-dihydroxy groups can rapidly chelate metal ions to form stable anthocyanin–metal complexes. However, it should be considered that the antioxidant properties of the polyphenolic compounds present in the extracts are generally difficult to attribute to a single compound since they can be the product of a synergistic effect between the majority and minority compounds [12].

In plants, the biosynthesis of phenolic compounds is mainly carried out by means of the shikimic acid route, in which the precursors are the amino acids phenylalanine and tyrosine [25]. The metabolism of phenylpropanoids comprises the complex branching of biochemical reactions that lead from L-phenylalanine to cumaroil CoA, a process that is initiated by the enzyme phenylalanine ammonoliase (PAL), which is regulated by the effect of light, and thanks to this route, biochemical compounds such as flavonoids, flavonols, anthocyanins, and tannins are synthesized [26].

In nature, there are seventeen knowns natural anthocyanidins, but only six of them are common in higher plants: cyanidin, peonidin, pelargonidin, malvidin, delphinidin, and petunidin [27], and their color depends on the number and orientation of the hydroxyl and methoxyl groups of the molecule, where increases in hydroxylation produce shifts to blue hues while increases in methoxylations produce red colorations [28]. These compounds have glycosidic substitutions at positions three and/or five with mono-, di-, or trisaccharides that increase their solubility. Another possible variation in the structure is the acylation of the sugar residues of the molecule with aliphatic or aromatic organic acids [28].

In the two extracts evaluated, many of the glycosidic substitutions of the anthocyanins corresponded to glucose and the organic acids that were acylating the sugar residues corresponded to the ferulic and acetic acids in the fruit and to the ferulic, malonic, pyruvic, and acidic acids p-coumaric in plants in vitro. On the other hand, the glycosidic substitutions of the phenolic compounds were more diverse in terms of the type of sugars, being formed by glucose, galactose, and soforose saccharides in the fruit, while in the leaves the saccharides present were glucose, xylose, rhamnose, rutin, and sambubiose.

In anthocyanins, acylation hinders the hydrolysis of the cationic red form of flavylium, allowing the formation of blue quinonoidal bases, however, acylated pigments retain more color at higher pH values than unmodified anthocyanins. According to Gras et al. [29], it is known that these phenolic acyl moieties form intramolecular complexes with the anthocyanidin nucleus that makes nucleophilic attack of the water at the C2 position of the pyrillium ring difficult, avoiding or partially delaying the formation of colorless hemicetals when increasing the pH. However, since these molecules are larger, they could have steric impediments to meet the active site of the enzymes evaluated, although they would be more stable at higher pH conditions.

The characterization of the anthocyanin profile of the fruit was consistent with that reported by Gironés-Vilaplana et al. [30] and by Brauch et al. [21] in extracts of *A. chilensis*, by Mazzuca et al. [31] in samples of purple lettuce and eggplant skin, Aaby et al. [32] in strawberry, Trikas et al. [33] in wine samples, and Zhang et al. [34] in onion skin, as well as in various Patagonian species of Chile found by Ruiz et al. [14], presenting glycosylated derivatives of delphinidin and cyanidin, compounds to which various therapeutic properties have been attributed [30,35,36,37,38].

Likewise, the phenolic compounds present in both extracts have been described by various authors in other plant species, such as Abu-Reidah et al. [39] in *Fragaria × ananassa* and *Annona cherimola*, Karim et al. [40] in *Theobroma cacao*, Bochi et al. [41] in *Dovyalis hebecarpa*; as well as Kumar et al. [42] in *Phyllanthus* sp., and Dartora et al. [43] in *Ilex paraguariensis*.

The flavonols kaempferol, quercetin, and myricetin, have identical chemical structures, except for the number of hydroxyl groups in ring B [44]. Glycosylated forms of these three compounds have been described in extracts of *A. chilensis* fruits by Brauch et al. [21]; Cespedes et al. [45], and Gironés-Vilaplana et al. [30].

Kaempferol is present in various fruits, vegetables, and medicinal plants such as grapes, strawberries, leeks, beans, cabbage, tea, broccoli, and moringa, and has been reported in in vivo studies to be capable of increasing insulin sensitivity and reducing the glycemic index [32,46,47].

Quercetin is present in plants such as onions, wheat, apples, blueberries, cherries, broccoli, grapes, leeks, lettuce, tomatoes, wild herbs, and citrus fruits and has been attributed antioxidant, anti-aging, anti-inflammatory, antiproliferative, anticancer, and cardioprotective properties [48].

Myricetin is a compound present in fruits, vegetables, tea, berries, and medicinal plants and has been reported to have antioxidant, anticancer, antimutagenic, cardioprotective, and antidiabetic activity, and it has also been found to reduce insulin resistance in type 2 diabetic rats [49].

On the other hand, phenolic acids are another group of compounds found in numerous plant species, which are chemically divided into two subgroups, namely: derivatives of hydroxybenzoic acids that have the C6–C1 structure, such as gallic acids, p-hydroxybenzoic, salicylic, vanillic, and ellagic, and derivatives of hydroxycinnamic acids that are aromatic compounds with a side chain of three carbons, among which are caffeic, ferulic, p-coumaric, and synaptic acids [50].

Ferulic acid has been shown to be an effective antioxidant in several in vitro tests and exhibits hydroxyl and peroxynitrite free radical scavenging properties [51]. Caffeic, chlorogenic, and caftaric acids have been shown to have powerful antioxidant properties, with greater antiradical activity of chlorogenic and caffeic acids compared with p-coumaric, which can be explained by the arrangement of the substituents in the molecule that favor the reactions with free radicals [52].

Particularly, derivatives of the Kaempferol compound were also reported in maqui fruit extracts by Gironés-Vilaplana et al. [30] and in strawberry extracts by Aaby et al. [32]. Furthermore, various reports have shown that kaempferol and some of their glycosides have antioxidant activity not only in vitro, but also in vivo due to the presence of a double bond at C2-C3 in conjugation with an oxo group at C4, and the presence of hydroxyl groups at positions C3, C5, and C4′ [53].

Some of the phenolic compounds identified in the evaluated aqueous extracts of *A. chilensis* were tested in the research carried out by Matsuda et al. [6] in the inhibition of the enzyme aldose reductase of rat lenses, obtaining the following IC_50_ values: Quercetin 2.2 µM, Rhamnetin 2.7 µM, Kaempferol 3-glucoside 5.1 µM, and Rutin 9.0 µM, while Fujita et al. [54] obtained an IC_50_ of 2.7 µM for the inhibition of recombinant human aldose reductase with Quercetin. Naeem et al. [55] also obtained an IC_50_ of 2.7 µM for the inhibition of recombinant human aldose reductase using Rhamnetin. Likewise, in the doctoral thesis carried out by Kraft [56], when testing the inhibition of the recombinant human aldose reductase enzyme, using methanolic extracts of *A. chilensis* fruits, an IC_50_ of 1.1 μg/mL was obtained in 50% aqueous methanol and an IC_50_ of 0.8 μg/mL in 100% methanol.

Chethan et al. [57] reported that quercetin isolated from *Eleusine coracana* extracts in methanol acidified with 1% HCl presented an IC_50_ of 25.23 μg/mL for the inhibitory activity of the enzyme aldose reductase purified from human lenses with cataracts. Kato et al. [58], using aqueous extracts of *Matricaria chamomilla* flowers, obtained an inhibition of the recombinant human aldose reductase enzyme with IC_50_ of 16.9 μg/mL.

El-Beshbishy and Bahashwan [59] tested aqueous extracts of *Ocimum basilicum* leaves, which exhibited a remarkable dose-dependent inhibition of the pig pancreatic alpha-amylase enzyme (IC_50_ = 42.50 mg/mL).

In the studies carried out by Ranilla et al. [60], the effect of aqueous extracts of dehydrated leaves of *Peumus boldus* in the inhibition of the activity of the porcine pancreatic alpha-amylase enzyme was evaluated, obtaining 85% inhibition enzymatic using 25 mg of DW from the extract.

Likewise, studies published by Rubilar et al. [61], using 50% ethanolic extracts, indicated that the fruit extract of *A. chilensis* presented a greater inhibition of the porcine pancreatic alpha-amylase enzyme with IC_50_ of 41.5 ± 3.6 mg/L, compared with the *Ugni molinae* stem extract with an IC_50_ of 56.6 ± 1.2 mg/L.

The results obtained in this investigation are promising since, on the one hand, there is no knowledge of other investigations where a model for obtaining *A. chilensis* tissues through in vitro culture has been developed to show efficacy in the inhibition of aldose reductase and pancreatic alpha-amylase enzymes. On the other hand, this represents a biotechnological opportunity for the elaboration and standardization of extracts that serve as raw material for formulations with possible pharmaceutical applications.

## 4. Materials and Methods

### 4.1. Preparation of Plant Extracts

The extracts were made at 10% fresh weight (*w*/*v*) at a concentration of 100 mg/mL from fruits and in vitro plants of *A. chilensis*, using aqueous infusions at 70 °C for 5 min of fresh tissues. Finally, the extracts were filtered and stored at −20 °C in darkness for later analysis.

Additionally, the aqueous preparation of standards for anthocyanin delphinidin-3-O-glucoside, cyanidin-3-O-glucoside, pelargonidin-3-O-glucoside, and peonidin-3-O-glucoside (Extrasynthese, Genay, France), as well as the phenolic compounds quercetin and ellagic acid (Sigma-Aldrich, St. Louis, MO, USA), at a concentration of 1 mM, was made to evaluate its content of total anthocyanins and phenols and the antioxidant and inhibitory activity of bovine aldose reductase enzyme.

### 4.2. Quantification of Total Anthocyanins by the Differential pH Method

The quantification of total anthocyanins was performed according to the protocol of Giusti and Wrolstad [62], including modifications for the TECAN microplate reader (Infinite M200 pro, Tecan, Switzerland). Using a 96-well plate, a first reaction was carried out with 50 μL of the extract and 250 μL of solution, pH 1.0 (0.025 M potassium chloride, Sigma-Aldrich, St. Louis, MO, USA). Separately, a second reaction with 50 μL of the extract and 250 μL of the solution, pH 4.5 (0.4 M sodium acetate, Sigma-Aldrich, MO, USA), for each of the prepared extracts, was carried out in triplicate. Subsequently, the reaction took place in the dark for 30 min using a TECAN microplate reader, while the absorbance at 520 nm and 700 nm was measured in both solutions. The total anthocyanin (TA) content was expressed in equivalent milligrams of anthocyanin per gram of tissue, using the following equations (Equations [1,2]):(1)$$A=\left(Abs.\;520nm-Abs.\;700nm\right)pH1.0-\left(Abs.\;520nm-Abs.\;700nm\right)pH4.5$$
(2)$$TA=\frac{A\times MW\times DF\times 1000}{\varepsilon \times l}$$
where *MW* is the molecular weight of anthocyanin, *DF* is the dilution factor of the extract, ε is the molar extinction coefficient of anthocyanin, and *l* is the pathlength in cm.

### 4.3. Determination of Total Phenols by the Folin–Ciocalteu Method

The total phenolic content was determined by the Folin–Ciocalteu method [63], for which a calibration curve for gallic acid (Sigma-Aldrich, St. Louis, MO, USA) was performed. Then, 115 µL of deionized water was placed in each well of a 96-well plate, and 5 µL of the different extracts obtained were added in two dilutions, with their respective targets. Subsequently, 20 µL of the Folin–Ciocalteu reagent (Sigma-Aldrich, St. Louis, MO, USA) was added. The mixture was incubated in the dark for 10 min, then 60 µL of 7% sodium carbonate was added and the analysis was performed on a TECAN microplate reader at 740 nm for 1 h. The results were expressed in milligrams of gallic acid equivalents per gram of dry weight (mg GAE/g DW).

### 4.4. Determination of Antioxidant Activity by DPPH Method

The antioxidant capacity by means of the DPPH free radical decolorization test was carried out according to the protocol proposed by Joyeux et al. [64], for which a 0.75 absorbance DPPH solution (Sigma-Aldrich, St. Louis, MO, USA) was prepared and approximately 5 µL of the extract and 195 µL of the DPPH solution was placed in each well and the measurement was made in the TECAN at 517 nm for 30 min. The antioxidant activity was expressed as IC_50_, which is the median inhibitory concentration, that is, the concentration of antioxidant compounds that could inhibit 50% of the DPPH radical.

### 4.5. Determination of the Anthocyanins and Phenolic Compounds Profile of the Extracts by LC-MS/MS

An LC-MS/MS analysis was performed for the different extracts obtained according to the protocol established at the Laboratory of Plant Physiology and Biotechnology of the University of Santiago de Chile [65], using an Agilent triple quadrupole mass spectrometer (MS/MS, 6400) equipped with an Agilent LC 1200 series (MS/MS, 6400; Agilent Technologies, Santa Clara, CA, USA). An RP-C18 column was used at flow rates of 1 mL min-1 at room temperature. The conditions for the analysis included a capillary voltage of 4000 V, a fogging pressure of 40 psi, and a drying gas temperature of 330 °C. The LC gradient was acetonitrile (Sigma-Aldrich, St. Louis, MO, USA) and 0.1% formic acid (Sigma-Aldrich, St. Louis, MO, USA). The anthocyanin analysis was performed using the positive ion mode, while the other phenolic compounds were analyzed in a negative mode.

### 4.6. Homogenization of Bovine Lenses

The homogenization of the bovine lenses was conducted according to the protocol of Del Corso et al. [66], with some modifications, for which 5 lenses were extracted by lateral incision of the bovine eye, which were washed with abundant cold distilled water to subsequently place them in a conical tube and quantify their volume, and then an equivalent volume of 50 mM potassium phosphate buffer (K_2_HPO_4_) pH 7.0 was added. The lenses were then homogenized in a Potter-Elvehjem homogenizer coupled to an electric drill until a milky-looking solution without visible lens fragments was obtained. Then, the homogenate was centrifuged at 13,000 rpm for 20 min and the supernatant was recovered to be immediately used to perform the aldose reductase tests.

### 4.7. Bovine Aldose Reductase Activity

The enzyme activity of bovine aldose reductase was performed according to the method of Fujita et al. [54], with some modifications. For this purpose, a reaction mixture containing 50 mM potassium phosphate buffer pH 7.0 was used with DL-glyceraldehyde (Sigma-Aldrich, St. Louis, MO, USA) as a substrate and the NADPH (Sigma-Aldrich, St. Louis, MO, USA) 150 µM cofactor and crude protein extract of the bovine lens homogenate in a total volume of 200 µL. The reaction mixture without the cofactor was incubated at 37 °C for 5 min, after which the reaction was initiated by adding the NADPH and immediately the decrease in absorbance at 340 nm every 30 s for 20 min was quantified using a reader of TECAN microplates. The decrease in absorbance was proportional to the cofactor’s oxidation.

### 4.8. Pancreatic Alpha-Amylase Activity

The enzyme activity of pancreatic alpha-amylase was performed according to the method of Rubilar et al. [61] with some modifications. For this purpose, porcine pancreatic alpha-amylase EC 3.2.1.1, type VI, (Sigma-Aldrich, St. Louis, MO, USA) was dissolved in ice-cold distilled water to a concentration of 3 U/mL. Potato starch (0.5%, *w*/*v*) in 20 mM phosphate buffer (pH 6.9) containing 6.7 mM sodium chloride was used as substrate. An aliquot of aqueous extracts of fruit and in vitro plants of *A. chilensis* (200 μL) and 400 μL of starch solution were mixed and preincubated for 5 min. The reaction was started by adding 200 μL of the enzyme solution; the reaction mixture was then incubated at 37 °C for 10 min. The reaction was stopped with 1.0 mL of dinitrosalicylic acid color reagent (Sigma-Aldrich, St. Louis, MO, USA) (96 mM 3,5-dinitrosalicylic acid in 0.4 M NaOH). The test tubes were incubated in a boiling water bath for 5 min and then cooled to room temperature. After the 1:4 dilution with deionized water, absorbance was measured at 540 nm using a reader of TECAN microplates. The same reaction mixture using Acarbose (Glucobay Bayer, Berlin, Germany) 50 mg/mL, without aqueous extracts, was used as a control.

### 4.9. Enzyme Inhibition

The enzyme activity was considered at 100% in the absence of the inhibitor, and the percentage of bovine aldose reductase and pancreatic alpha-amylase enzyme inhibition was determined after subtracting the blank value with an average of at least three replicates by the following equation:(3)$$\%\;Inhibition=\frac{\left(∆Abs.\;sample\;with\;inhibitor-∆Abs.\;blank\;of\;sample\right)}{∆Abs.\;control-∆Abs.\;blank\;control}\times 100$$

### 4.10. Statistical Treatment

Statistical treatments were performed with the STATA 14 Special Edition program using a 95% confidence interval, and the level of significance corresponds to *t* < 0.05. It was determined that as a null hypothesis (Ho) the means to be compared were statistically equal, while the alternative hypothesis (H1) corresponds to the means to be compared being statistically different.

## Figures and Tables

**Figure 1 plants-11-02772-f001:**
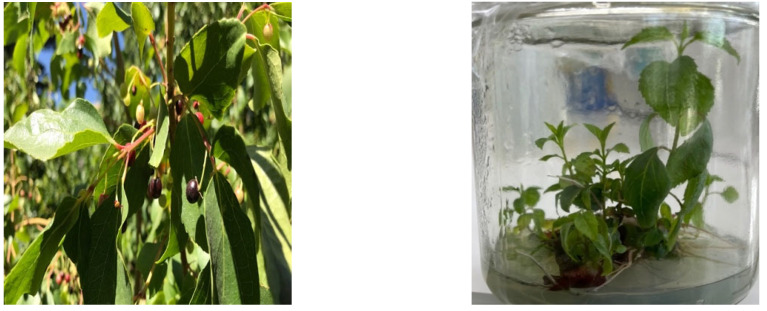
Fruits and in vitro plants of *Aristotelia chilensis* (Maqui).

**Figure 2 plants-11-02772-f002:**
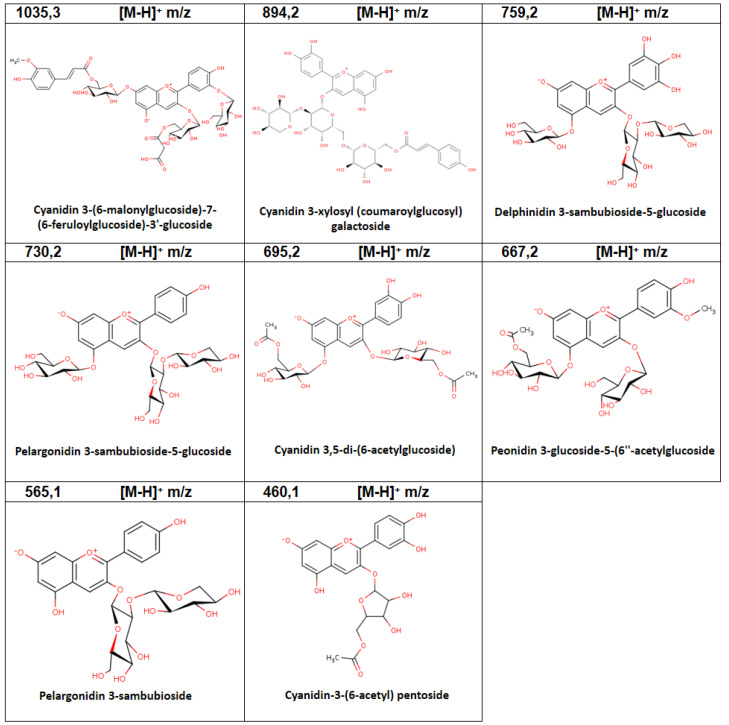
Chemical structure of the main anthocyanins identified by LC-MS/MS analysis for the aqueous extracts of fruit of *A. chilensis*.

**Figure 3 plants-11-02772-f003:**
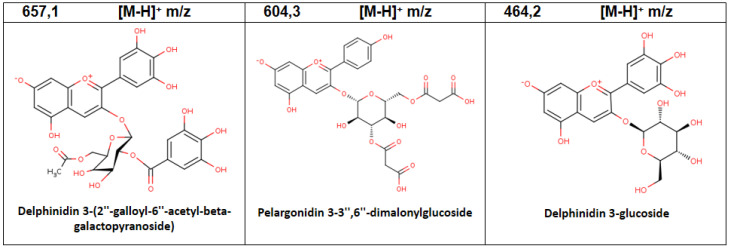
Chemical structure of the anthocyanins identified by LC-MS/MS analysis for the aqueous extracts of in vitro plant of *A. chilensis*.

**Figure 4 plants-11-02772-f004:**
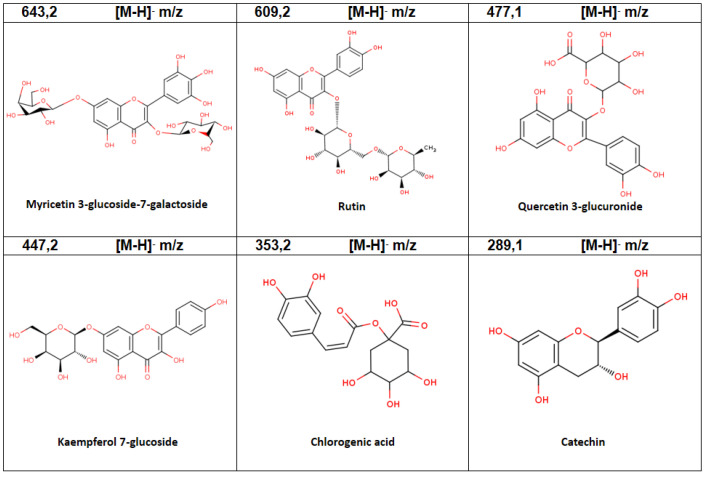
Chemical structures of phenolic compounds identified by LC-MS/MS analysis of aqueous extracts of fruits of *A. chilensis*.

**Figure 5 plants-11-02772-f005:**
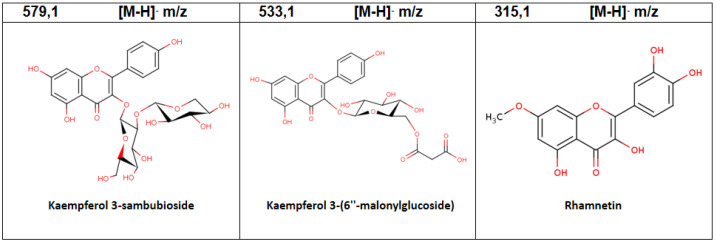
Phenolic compounds identified by LC-MS/MS analysis of aqueous extracts of in vitro plants of *A. chilensis*.

**Table 1 plants-11-02772-t001:** Content of total anthocyanins, total phenols, DPPH (IC_50_) of aqueous extracts of fruit and in vitro plant of *A. chilensis*. Mean ± standard deviation. Significant differences are denoted by different letters, (*p* < 0.05).

Extract	Total Anthocyanin	Total Phenol	DPPH (IC_50_)
	mg Anthocyanin/g DW	mg GAE/g DW	mg/g DW
Fruit	7.63 ± 0.09 ^a^	79.90 ± 3.77 ^c^	0.11 ± 0.01 ^e^
In vitro plant	0.40 ± 0.01 ^b^	19.65 ± 2.06 ^d^	0.12 ± 0.01 ^e^

**Table 2 plants-11-02772-t002:** Anthocyanins identified by HPLC-MS/MS analysis for aqueous extracts of fruit and in vitro plant of *A. chilensis*.

Peak	RT	Molecule	Chemical Formula	[M-H]^+^ *m*/*z*	Relative Abundance (%)
1	0.74	Cyanidin 3-(3″,6″-dimalonylglucoside)	C_27_H_24_O_17_	621.1	9.71
2	0.86	Pelargonidin 3-glucoside	C_21_H_20_O_10_	429.1	1.06
3	1.09	Delphinidin 3,5-diglucoside	C_27_H_30_O_17_	627.1	20.53
4	1.39	Cyanidin-3-(6-acetyl) pentoside	C_22_H_20_O_11_	460.1	1.20
5	2.86	Delphinidin 3-(2″-galloylgalactoside)	C_28_H_24_O_16_	617.1	8.58
6	3.44	Delphinidin 3,7-diglucoside-3′,5′-di (6-p-coumaroyl-beta-glucoside)	C_57_H_62_O_31_	1242.3	0.68
7	3.75	Pelargonidin 3-sambubioside	C_26_H_28_O_14_	565.1	0.23
8	4.04	Delphinidin 3-sambubioside-5-glucoside	C_32_H_38_O_21_	759.2	2.00
9	5.23	Peonidin 3-glucoside-5- (6″-acetylglucoside)	C_30_H_34_O_17_	667.2	0.70
10	7.54	Peonidin 3-rutinoside	C_28_H_32_O_15_	607.2	3.41
11	8.21	Cyanidin 3-(6-malonylglucoside)-7- (6-feruloylglucoside)-3′-glucoside	C_46_H_50_O_27_	1035.3	1.86
12	9.03	Cyanidin 3-xylosyl (coumaroylglucosyl) galactoside	C_41_H_45_O_22_	894.2	4.33
13	9.48	Cyanidin 3-sambubioside	C_26_H_29_O_15_	587.2	1.71
14	10.06	Malvidin 3-rutinoside	C_29_H_34_O_16_	638.2	7.21
15	10.32	Cyanidin-3-(2′-acetylrutinoside)	C_29_H_32_O_31_	635.2	5.53
16	11.08	Cyanidin 3,5-di-(6-acetylglucoside)	C_31_H_34_O_18_	695.2	0.97
17	11.29	Cyanidin 3-[6-(6-p-coumarylglucosyl)- 2-xylosylgalactoside]	C_41_H_14_O_22_	887.2	1.16
18	11.51	Pelargonidin 3-sambubioside-5-glucoside	C_32_H_38_O_19_	730.2	0.72
19	11.84	Pelargonidin 3-(6″-p-coumaryl sambubioside)-5-(6″′-malonylglucoside)	C_44_H_46_O_24_	958.2	1.45
20	12.10	Delphinidin-3-(6-feruloyl)-5-diglucoside	C_27_H_50_O_27_	803.1	26.95

**Table 3 plants-11-02772-t003:** Anthocyanins identified by LC-MS/MS analysis for aqueous extracts of in vitro plant of *A. chilensis*.

Peak	RT	Molecule	Chemical Formula	[M-H]^+^ *m*/*z*	Relative Abundance (%)
1	0.83	Pelargonidin 3-glucoside	C_21_H_20_O_10_	429.1	10.27
2	1.01	Delphinidin 3-glucoside	C_21_H_20_O_12_	464.2	14.32
3	1.38	Cyanidin 3-(3″,6″-dimalonylglucoside)	C_27_H_24_O_17_	621.1	2.84
4	1.66	Cyanidin 3-sambubioside	C_26_H_29_O_15_	587.2	2.73
5	2.39	Delphinidin 3,7-diglucoside-3′,5′- di(6-p-coumaroyl-beta-glucoside)	C_57_H_62_O_31_	1242.3	0.47
6	2.73	Pelargonidin 3-3″,6″-dimalonylglucoside	C_27_H_42_O_16_	604.3	1.31
7	4.17	Cyanidin-3-(2′-acetylrutinoside)	C_29_H_32_O_16_	635.2	2.50
8	4.49	Delphinidin 3,5-diglucoside	C_27_H_30_O_17_	627.1	0.91
9	6.83	Delphinidin 3-(2″-galloyl-6″-acetyl-beta-galactopyranoside)	C_30_H_26_O_17_	657.1	3.04
10	7.32	Pelargonidin 3-(6″-p-coumaryl sambubioside)-5-(6″′-malonylglucoside)	C_44_H_46_O_24_	958.2	0.80
11	7.52	Peonidin 3-rutinoside	C_28_H_32_O_15_	607.2	5.50
12	8.26	Malvidin 3-rutinoside	C_29_H_34_O_16_	638.2	0.51
13	8.88	Delphinidin 3-(2″-galloylgalactoside)	C_28_H_24_O_16_	617.1	1.61
14	9.51	Cyanidin 3-[6-(6-p-coumarylglucosyl)- 2-xylosylgalactoside]	C_41_H_44_O_22_	887.2	0.94
15	12.09	Delphinidin-3-(6-feruloyl)-5-diglucoside	C_46_H_50_O_27_	803.1	52.24

**Table 4 plants-11-02772-t004:** Phenolic compounds identified by LC-MS/MS analysis of aqueous extracts of fruits of *A. chilensis*.

Peak	RT	Molecule	Chemical Formula	[M-H]^−^ *m*/*z*	Relative Abundance (%)
1	0.74	Catechin	C_15_H_14_O_6_	289.1	2.53
2	1.13	Quercetin 4′-galactoside	C_20_H_18_O_12_	451.1	35.90
3	1.67	Chlorogenic acid	C_16_H_18_O_9_	353.2	2.68
4	2.93	3,5-di-chlorogenic acid	C_25_H_24_O_12_	515.2	22.42
5	3.47	Myricetin 3-glucoside-7-galactoside	C_27_H_30_O_18_	643.2	3.68
6	4.05	Kaempferol 3-[2″-glucosyl-6″-acetyl-galactoside] 7-glucoside	C_27_H_30_O_19_	813.1	3.51
7	6.18	Tetramethylquercetin 3-rutinoside	C_31_H_38_O_16_	666.2	3.25
8	7.67	Granatin B	C_41_H_28_O_27_	951.1	1.51
9	8.52	Rutin	C_27_H_30_O_16_	609.2	1.14
10	9.06	Quercetin 3-glucuronide	C_21_H_18_O_13_	477.1	0.95
11	10.79	Quercetin 3-(6″″-ferulylsophorotrioside)	C_43_H_48_O_25_	965.3	1.48
12	11.88	Kaempferol 3-(4″,6″-diacetylglucoside)- 7-rhamnoside	C_31_H_34_O_17_	679.2	6.34
13	13.30	Kaempferol 7-glucoside	C_21_H_20_O_11_	447.2	14.61

**Table 5 plants-11-02772-t005:** Phenolic compounds identified by LC-MS/MS analysis of aqueous extracts of in vitro plants of *A. chilensis*.

Peak	RT	Molecule	Chemical Formula	[M-H]^−^ *m*/*z*	Relative Abundance (%)
1	0.83	3,5-di-chlorogenic acid	C_25_H_24_O_12_	515.2	6.11
2	1.11	Quercetin 4′-galactoside	C_20_H_18_O_12_	451.1	15.94
3	1.36	Kaempferol 3-(6″-malonylglucoside)	C_24_H_22_O_14_	533.1	7.65
4	1.69	Rhamnetin	C_16_H_12_O_7_	315.1	12.16
5	3.70	Tetramethylquercetin 3-rutinoside	C_31_H_38_O_16_	666.2	1.93
6	4.33	Kaempferol 3-[2″-glucosyl-6″-acetyl-galactoside] 7-glucoside	C_27_H_30_O_19_	813.1	1.29
7	4.57	Kaempferol 3-sambubioside	C_26_H_28_O_15_	579.1	5.59
8	7.33	Quercetin 3-(6″″-ferulylsophorotrioside)	C_43_H_48_O_25_	965.3	2.78
9	7.98	Granatin B	C_41_H_2_8O_27_	951.1	11.17
10	9.90	Kaempferol 3-(4″,6″-diacetylglucoside)- 7-rhamnoside	C_31_H_34_O_17_	679.2	35.38

**Table 6 plants-11-02772-t006:** Inhibition of bovine aldose reductase (AR) and pancreatic alpha-amylase (AA) enzymes for aqueous extracts of fruit and in vitro plant of *A. chilensis* and aqueous solutions of the anthocyanins and phenolic compounds standards. Mean ± standard deviation.

Inhibitor	AR % Inhibition	AA % Inhibition
Fruit	85.54 ± 1.86 ^a^	29.64 ± 0.63 ^a^
In vitro plant	75.67 ± 1.21 ^b^	47.66 ± 0.66 ^b^
Delphinidin-3-O-glucoside	93.45 ± 5.67 ^c^	73.01 ± 0.05 ^c^
Cyanidin-3-O-glucoside	83.52 ± 1.98 ^a^	51.96 ± 0.22 ^d^
Pelargonidin-3-O-glucoside	77.33 ± 1.26 ^b^	44.56 ± 1.37 ^b^
Peonidin-3-O-glucoside	44.97 ± 1.61 ^d^	54.92 ± 1.42 ^d^
Quercetin	74.47 ± 1.78 ^b^	29.96 ± 5.27 ^a^
Ellagic acid	93.97 ± 2.92 ^c^	15.16 ± 0.27 ^e^
Acarbose	NT	96.70 ± 0.26 ^f^

NT: not tested. Significant differences are denoted by different letters, (*p* < 0.05).

## Data Availability

The data presented in this study are available on request from the corresponding author.

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
