# Peer review of "Extracts of Fruits and Plants Cultivated In Vitro of Aristotelia chilensis (Mol.) Stuntz Show Inhibitory Activity of Aldose Reductase and Pancreatic Alpha-Amylase Enzymes"

_plants, 2022, doi:10.3390/plants11202772_

Round 1
Reviewer 1 Report
The article entitled " Extracts of fruits and plants cultivated in vitro of Aristotelia chilensis (Mol.) Stuntz, show inhibitory activity of the enzymes related to diabetes, Aldose reductase and Pancreatic alpha-amylase" presented to me for review is original work. Despite the fact that the work is probably interesting for biotechnology circles, it is difficult to find any novelties in it. The authors present the previously obtained culture of this specie. The title is a bit too long for me and I think it should be corrected. There is no need to exchange research, just add "and biological properties"-please rephrase. Introduction and abstract are written correctly, you can quite clearly understand the purpose of the work but I suggest to add several information about plant (family, biological properties etc..). The results are also described quite clearly and legibly, thanks to which the recipient has a clear view of the next stages of the study. The material and methods section is described quite modestly but all necessary things are included. The material and methods section is described quite modestly but all necessary things are included. the discussion, on the other hand, is unfortunately quite laconic and inadequate to the part of the work. Apart from a small selection of the literature cited by the authors, there is no discussion of the results of biological properties obtained in the course of the research. Therefore, it is imperative that this part of the work be done correctly. Besides, the discussion is endless ... lack of proper ending and summary in the form of conclusions that should define the overall nature of the work- please correct and add citations in the first part of the discussion. Please correct some punctuation and stylistic errors in the manuscript.
Reviewer 2 Report
In this work, authors report on extracts from fruits and plants that exhibit inhibitory activity of enzymes related to diabetes, such as aldose reductase and pancreatic alpha-amylase. The paper is interesting, a few typos, unnecessary gaps and English mistakes could be found in the manuscript. Overall, the results are supported by the data presented. I have a few minor comments in order for the authors to improve further the quality of this manuscript.
· A few typos, unnecessary gaps or English mistakes need to be corrected by the authors. They should check the whole manuscript carefully. For example, in vitro should be everywhere in the text in italics, please correct it on page 6 line 170-171, page 8 line 190, page 9 line 222, page 10 line 265. Grammar mistakes should be corrected, for example delete it from page 2 line 81. Unnecessary gaps page 2 line 57, page 10 line 275, page 11 line 302, page 11 line 337. Please delete parenthesis before EC page 12, line 379.
· The abstract looks more like a mini version of the introduction rather than a proper abstract. I think the first two paragraphs should be deleted and replaced by a few sentences about the problem and the solution that involves the two enzymes.
· I think in the keywords the two enzymes should be mentioned, maybe replacing diabetes.
· In the materials and methods please specify from where you purchased the following chemicals, sodium acetate, potassium chloride, sodium carbonate, acetonitrile, formic acid, potassium phosphate buffer and phosphate buffer you mention later on, in case it is with sodium.
· On page 3 in line 118 the relevant reference is missing.
· I think the authors need to replace their references in the text with numbers rather than using the name of the first author of each reference.
Author Response
Dear Reviewer
Our answers are in the attached document.

Reviewer 3 Report
Please report the abbreviations in extenso at their first appearance in the text.
Please include decimal points as regards the m/z ratios in tables 2-5 and figures 2-5.
In tables 2-5 please substitute comma with fullstop.
Authors should improve the description of LC-MS conditions.
Chromatograms have to be included also as supplementary material.
Author Response

(The authors gave the same response as above.)

Reviewer 4 Report
The manuscript entitled Extracts of fruits and plants cultivated in vitro of Aristotelia chilensis (Mol.) Stuntz, show inhibitory activity of the enzymes related to diabetes, Aldose reductase and Pancreatic alpha-amylase. The authors definitely evaluated the extracts of aqueous infusions of fruits and in vitro plants of A. chilensis, particularly in relation to their antioxidant activity and inhibitory effects on Bovine aldose reductase and Pancreatic alpha amylase enzymes.
I think this is a useful article and I think it makes a potent scientific contribution to plant resource. I would recommend the manuscript for major revision.
Just major points,
The introduction must be improved by incorporating more recent references including diabetes and antioxidants.
Add SOD data of Fruit and invitro plant in Table 1.
In Table 2, please add spectrum data of anthocyanins identified by anthocyanins identified by HPLC-MS/MS analysis in supplementary.
In Table 3, please add spectrum data of anthocyanins identified by LC-MS/MS analysis in supplementary.
In Table 4, please add spectrum data of anthocyanins identified by phenolic compounds indentified by LC-MS/MS analysis in supplementary.
In Table 5, please add spectrum data of anthocyanins identified by Phenolic compounds indentified by LC-MS/MS analysis in supplementary.
In conclusion, please the contents detailed should be addressed including future scope and prospects in feasibility of inhibitory activity of the enzymes related to diabetes, aldose reductase and pancreatic alpha amylase.
Author Response

(The authors gave the same response as above.)

Round 2
Reviewer 1 Report
The authors revised the manuscript as suggested by the Reviewer.
Reviewer 4 Report
The authors have improved the manuscript for publication.